# Arpin Regulates Migration Persistence by Interacting with Both Tankyrases and the Arp2/3 Complex

**DOI:** 10.3390/ijms22084115

**Published:** 2021-04-16

**Authors:** Gleb Simanov, Irene Dang, Artem I. Fokin, Ksenia Oguievetskaia, Valérie Campanacci, Jacqueline Cherfils, Alexis M. Gautreau

**Affiliations:** 1CNRS UMR7654, Institut Polytechnique de Paris, 91120 Palaiseau, France; gleb.simanov@polytechnique.edu (G.S.); idang@mail.med.upenn.edu (I.D.); artem.fokin@polytechnique.edu (A.I.F.); oguieve@gmail.com (K.O.); 2Laboratoire d’Enzymologie et Biochimie Structurales, CNRS, 91190 Gif-sur-Yvette, France; valerie.campanacci@i2bc.paris-saclay.fr (V.C.); jacqueline.cherfils@ens-paris-saclay.fr (J.C.)

**Keywords:** cell migration, migration persistence, Arpin, Tankyrase, Arp2/3

## Abstract

During cell migration, protrusion of the leading edge is driven by the polymerization of Arp2/3-dependent branched actin networks. Migration persistence is negatively regulated by the Arp2/3 inhibitory protein Arpin. To better understand Arpin regulation in the cell, we looked for its interacting partners and identified both Tankyrase 1 and 2 (TNKS) using a yeast two-hybrid screening and coimmunoprecipitation with full-length Arpin as bait. Arpin interacts with ankyrin repeats of TNKS through a C-terminal-binding site on its acidic tail, which overlaps with the Arp2/3-binding site. Arpin was found to dissolve the liquid–liquid phase separation of TNKS upon overexpression. To uncouple the interactions of Arpin with TNKS and Arp2/3, we introduced point mutations in the Arpin tail and attempted to rescue the increased migration persistence of the Arpin knockout cells using random plasmid integration or compensating knock-ins at the *ARPIN* locus. Arpin mutations impairing interactions with either Arp2/3 or TNKS were insufficient to fully abolish Arpin activity. Only the mutation that affected both interactions rendered Arpin completely inactive, suggesting the existence of two independent pathways, whereby Arpin controls the migration persistence.

## 1. Introduction

Cell migration depends on various types of membrane protrusions. Most membrane protrusions are driven by cortical actin polymerization [1]. In migrating cells, adherent membrane protrusions at the leading edge fuel the cell movement. Actin networks at the leading edge are branched by the Arp2/3 complex that nucleates new actin filaments from the side of pre-existing ones. Arp2/3 activation in membrane protrusions is under the control of the WAVE complex, which, in turn, is regulated by the small GTPase Rac1 [2,3].

Arpin was identified as an Arp2/3 inhibitory protein that antagonizes WAVE activity [4]. Arpin is composed of a folded domain and of a C-terminal acidic tail extending from this core [5]. Through its acidic tail, Arpin competes with the nucleation-promoting factors (NPFs), such as WAVE [4]. Arpin thus acts earlier than other Arp2/3 inhibitory proteins, such as coronins or GMFs, which remove Arp2/3 from junctions of branched actin networks [3].

Migration persistence is the result of feedback loops that sustain Rac1 activation in the cell regions, where Rac1 earlier induced the formation of branched actin [6,7]. By inhibiting Arp2/3, Arpin interrupts this positive feedback and decreases the migration persistence, thus allowing the cells to pause and change direction [4,8].

To better understand how Arpin is regulated in the cell, we searched for additional Arpin interacting partners and identified Tankyrase 1 and 2 (TNKS) as major Arpin partners. TNKS are pleiotropic regulators of many cell functions through binding, modification, and the downregulation of various proteins. However, TNKS did not appear to regulate Arpin’s Arp2/3 inhibitory function but, rather, to serve as another Arpin effector in the regulation of migration persistence.

## 2. Results

### 2.1. Arpin Binds to Tankyrase 1 and 2

To identify the proteins that bind to Arpin, we first immunoprecipitated Arpin from a 293-cell line stably expressing a Protein C (PC)-tagged version of Arpin through its epitope tag. Arpin was efficiently immunoprecipitated from the lysate prepared from cells stably expressing tagged Arpin but not from a control cell line transfected with the empty plasmid. Silver staining was used to identify the potential interacting partners (Figure 1A). In single-step immunoprecipitation, many bands were detected, including immunoglobulin light and heavy chains. However, two specific bands between the 120- and 150-kDa markers were clearly detected when Arpin was immunoprecipitated but not in the control lane. These two proteins were identified by LC-MS/MS as Tankyrase1 (TNKS1, 142 kDa) and Tankyrase2 (TNKS2, 127 kDa). With the expectation of identifying additional potential interacting partners of the Arpin protein, we performed a yeast two-hybrid screening of a library containing random primed human placenta cDNAs, with full-length Arpin as bait. More than 10^8^ clones were analyzed by yeast mating. Out of the 187 clones selected, 177 corresponded to either TNKS1 or TNKS2 (Figure 1B). The remaining 10 clones were comparatively of low confidence, as they corresponded to out-of-frame fusions or to DNA sequences that were not annotated as protein-encoding genes. These two approaches thus point at TNKS as major Arpin-interacting partners.

Both TNKS are composed of three regions. In order from the N- to C-terminus, these proteins contain ankyrin repeats organized into Ankyrin Repeat Clusters (ARCs), a Sterile Alpha Motif (SAM), which mediates oligomerization, and a C-terminal Poly ADP Ribosyl Polymerase (PARP) catalytic domain [9]. Since all yeast two-hybrid clones interacting with full-length Arpin are mapped to the N-terminal region composed of ARCs, we produced and purified full-length Arpin and the ARC4 of TNKS2 (Appendix A), which had been previously crystallized [10]. When the two proteins were mixed, a new molecular species was detected by Size Exclusion Chromatography–Multi-Angle Light Scattering (SEC-MALS). This species displayed a mass corresponding to a 1:1 complex (Figure 1C).

### 2.2. TNKS Do Not Regulate Arpin Levels

TNKS are pleiotropic regulators of various cellular functions, including telomere maintenance, mitosis regulation, Wnt signaling, insulin-dependent glucose uptake, and the Hippo–YAP pathway [11,12,13,14,15,16,17]. These multiple TNKS functions usually require binding to substrates through ARCs and poly ADP ribosylation, also called PARylation, through the catalytic PARP domain [18]. In most cases, PARylated proteins are ubiquitinated by the E3 ubiquitin ligase RNF146, which recognizes the poly-ADP ribose chain through its WWE domain, and then degraded by proteasomes [19].

The TNKS turnover is fast, because they PARylate themselves. To investigate whether the Arpin turnover occurs through a similar mechanism, we blocked TNKS catalytic activity with the XAV939 inhibitor [16]. As expected, this treatment resulted in increased levels of both TNKS and of their substrate Axin1. The Arpin levels were, however, not affected by the XAV939 treatment (Figure 2A). This observation is in line with proteomics analyses of the TNKS function: Levels of Arpin (referred to in these large-scale studies as C15ORF38) were also found to be unchanged in TNKS double-knockout versus control 293T cells [20], and the Arpin–TNKS interaction was unaffected irrespective of whether the TNKS PARylation activity was blocked by XAV939 [21]. We nonetheless attempted to detect the potential PARylation of Arpin. To this end, we used the WWE domain of E3 ligase RNF146 (Appendix A) to pull down the PARylated proteins [19]. In the WWE pulldown, TNKS1, but not Arpin, was retrieved (Figure 2B).

Together, these experiments indicated that Arpin is a direct TNKS partner that is unlikely to be subjected to PARylation-mediated degradation. This is different from the majority of TNKS-binding partners; however, several other TNKS-interacting partners that are not PARylated have been previously identified, such as Mcl-1L, GDP Mannose 4,6 Dehydratase, CD2AP, and SSSCA1 [22,23,24,25].

### 2.3. Arpin Binds to TNKS via Its C-Terminal Acidic Tail

ARCs recognize a consensus motif, the octapeptide RXXXXGXX, defined through the screening of a peptide library [10]. Arpin contains three putative TNKS-binding sites, which were examined by substituting the required G residue by A at positions 8, 189, and 218. We expressed PC-tagged Arpin mutant forms in 293T cells and found that the G218 residue is the only critical one for Arpin association with TNKS (Figure 2C). This TNKS-binding motif is located within the acidic tail of Arpin and overlaps with the previously described Arp2/3 interaction site (Figure 2D) [4]. We therefore investigated a possible competition between the Arp2/3 complex and TNKS binding. GST pulldown with Arpin in lysates from mouse embryonic fibroblasts (MEF) retrieved both TNKS and the Arp2/3 complex (Figure 2E). Adding an excess of purified ARC4 displaced not only TNKS but, also, the Arp2/3 complex, due to overlapping binding sites. In order to understand whether TNKS could influence Arpin–Arp2/3 interactions in cells, we immunoprecipitated endogenous Arpin from 293T wild type and TNKS double-knockout (KO) cells [20]. The same amount of Arp2/3 complex coprecipitated with Arpin irrespective of the presence of TNKS (Figure 2F). Thus, the Arpin–TNKS interaction does not appear to modulate the Arpin–Arp2/3 interaction in vivo. Since TNKS does not regulate the levels of Arpin, nor its Arp2/3 inhibitory function, we then investigated what might be the possible outcome of the Arpin–TNKS interaction.

We attempted to uncouple TNKS and Arp2/3 binding using mutations in the Arpin acidic tail. To prevent TNKS binding, we replaced R213 and G218 with A or D residues. Alanine substitutions are the most widely used ones to impair binding sites, but introducing aspartate increases the negative charge of the Arpin tail, a requisite for Arp2/3 binding [26]. To impair Arp2/3 binding, we substituted the conserved C-terminal tryptophan of Arpin with alanine (W224A). We expressed these mutant forms of Arpin in 293T cells and immunoprecipitated them to analyze their binding partners (Figure 2G). As expected, the TNKS interaction was undetectable when R213 or G218 of Arpin was mutated. The Arp2/3 interaction was below the detection limit with the W224A substitution. It was also surprisingly affected by the G218D substitution, even though this mutation increased the overall acidity of the tail. G218D thus impaired both TNKS and Arp2/3 binding.

### 2.4. Arpin Controls the Ability of TNKS to Form Biomolecular Condensates

TNKS can homo- and hetero-oligomerize via their SAM motif [27]. The polymeric state reinforces the PARP activity of TNKS and is required for Wnt signaling [27,28]. Furthermore, when overexpressed, TNKS forms cytosolic aggregates [21,27,28,29]. We found that the aggregates formed by GFP-TNKS2 in transiently transfected MCF10A cells (Figure 3A) were dynamic, since they could fuse (Figure 3B) and quickly recover their fluorescence after photobleaching (Figure 3C). TNKS2 aggregates recovered up to 70% of their fluorescence intensity within 100 s. These properties suggested that GFP-TNKS2 aggregates underwent liquid–liquid phase separation and formed so-called biomolecular condensates [30]. Such condensates are often controlled by multivalent interactions. TNKS display five ARCs for each protomer of a multimer. The TNKS partners that display several binding sites, such as Axin [31], can bring together several multimeric TNKS units and may subsequently promote a liquid–liquid phase transition.

We thus examined whether Arpin, which possesses a single binding site, might regulate the formation of biomolecular condensates by TNKS. As a negative control, we used the G218D mutant form, which displays reduced Arp2/3 binding, in addition to impaired TNKS interactions. When we co-expressed Arpin with GFP-TNKS2, wild-type Arpin, but not G218D Arpin, prevented the formation of TNKS condensates (Figure 3D). This behavior was different from the ones of endogenous or overexpressed Axin, which were reported to colocalize with TNKS condensates [27,28]. The co-expression of wild-type Arpin decreased the fraction of aggregate-positive cells (Figure 3E) and the number of condensates per cell (Figure 3F). Since these results were obtained in the context of TNKS overexpression, we also examined whether Arpin would exert this function on endogenous TNKS. In MCF10A cells, TNKS were diffusely localized in the cytoplasm of WT and *ARPIN* KO cells (Appendix A). Upon XAV939 treatment, TNKS exhibited condensation. No difference in TNKS condensation was observed between WT and *ARPIN* KO cells. Since the ability to form biomolecular condensates is thought to correspond to increased PARylation activity, we also examined the levels of TNKS and of their substrates Axin1 and PTEN. However, these levels were unchanged in *ARPIN* KO cells compared to parental cells (Appendix A). In conclusion, Arpin has a striking role in preventing TNKS from forming biomolecular condensates when both components are overexpressed, but the implications of this result at the endogenous level of expression are not clear.

### 2.5. The Arpin–TNKS Interaction Participates in the Regulation of Cell Migration

To examine the role of the Arpin–TNKS interaction, we used the MCF10A *ARPIN* KO cell line, previously generated using CRISPR/Cas9 [32], to isolate clones stably re-expressing exogenous WT Arpin or its derivatives. We first focused on two Arpin mutations, G218A and W224A, that impair TNKS and Arp2/3 binding, respectively (Figure 2G). Exogenous Flag-tagged Arpins were moderately overexpressed compared to the endogenous Arpin in the parental MCF10A cell line (Figure 4A). A major role of the Arp2/3 pathway in cell migration is to mediate migration persistence through positive feedback, and the Arp2/3 inhibitory protein Arpin antagonizes this role [4,6]. We recorded the random migration of single cells using these cell lines, extracted the cell trajectories, and analyzed them using DiPer software [33]. As previously reported [32], *ARPIN* KO cells exhibited higher migration persistence compared to parental MCF10A cells. The expression of wild-type Arpin fully rescued the *ARPIN* KO phenotype (Figure 4B). The W224A mutant form, whose interaction with Arp2/3 was impaired, partially rescued the phenotype almost as efficiently as wild-type Arpin. The G218A mutant form, which still binds to Arp2/3 but not to TNKS, also partially rescued the phenotype but less efficiently than W224A. All the migration parameters extracted from the cell trajectories are displayed in Appendix A for reference, but the only parameter that was regulated by Arpin in all the cell systems was migration persistence, not speed or the mean square displacement [4,6,32]. These results suggested that the Arpin–TNKS interaction could regulate the migration persistence, but we sought to confirm them in cell clones where Arpin was not overexpressed.

For this purpose, we designed a GFP-Arpin Knock-In (KI) strategy. Briefly, we introduced two double-stranded breaks (DSBs) using CRISPR/Cas9 to excise the exons encoding the *ARPIN* open reading frame from *ARPIN* KO MCF10A cells and provided a donor plasmid encoding either GFP-Arpin WT, G218A, G218D, or W224A for Homology-Directed Repair (HDR; Figure 5A and the Methods section). GFP-Arpin expression was confirmed by Western blotting in stable clones isolated upon puromycin selection (Figure 5B). Indeed, transgene expression was, overall, at the endogenous level, even though differences could still be observed between the constructs and clones. Due to low GFP-Arpin expression levels, we performed immunofluorescence staining of fixed cells to enhance the signal. All forms of GFP-Arpin were distributed throughout the cytoplasm and the nucleus, similar to endogenous Arpin (Appendix A).

We then performed the single-cell migration assay on the obtained KI clones. WT Arpin fully rescued the *ARPIN* KO phenotype (Figure 5C). Near-total rescue was also observed in the case of W224A Arpin. The phenotypical variation between the clones was not an effect of the expression levels. The G218A Arpin provided a partial rescue that failed to reach significance, indicating that this mutation that abolished TNKS binding had a greater effect than W224A in the KI, as well as in the overexpression system. Only G218D Arpin with an impaired interaction with both Arp2/3 and TNKS was completely unable to rescue the *ARPIN* KO phenotype. All the migration parameters extracted from the cell trajectories are displayed in Appendix A for reference. These results suggest that the interactions of Arpin with Arp2/3 and TNKS represent two independent pathways that both regulate cell migration.

## 3. Discussion

Here, we report that TNKS are major Arpin-interacting partners in the cell. TNKS bind to Arpin’s exposed C-terminal tail that protrudes from a folded core domain [5]. The C-terminal tail carries the previously reported Arp2/3 inhibitory binding site [4]. The TNKS-binding site overlaps with the Arp2/3-binding site within the Arpin tail, and we found that one ARC of TNKS can displace the Arp2/3 bound to the tail of Arpin in vitro. However, this competition does not appear to occur in vivo, since the amount of Arp2/3 bound to Arpin does not increase in TNKS double-KO cells. Arpin is thought to bind Arp2/3 at the lamellipodial edge in response to Rac1 signaling. The results obtained here rather suggest that Arpin binds to TNKS in a diffuse manner in the cytosol or the nucleus. In the cell, the lack of competition of Arp2/3 and TNKS for Arpin binding might be due to the fact that these two partners bind Arpin in different locations.

Our analysis of the point mutations of the Arpin tail suggests that Arp2/3 and TNKS are both important for the regulation of migration persistence. Indeed, point mutations that specifically impair binding to either of the Arpin partners display only a partial loss of activity, even in a clean KI context with endogenous levels of expression. In contrast, the G218D mutation that significantly impairs binding to both Arp2/3 and TNKS displayed a clear loss of function. Previously, we reported that the deletion of the whole C-terminal tail fully inactivated Arpin [4]. This is consistent with our current results, but it can no longer be attributed solely to the abolishment of Arp2/3 binding. TNKS binding to Arpin participates in the regulation of migration persistence independently of Arp2/3 binding (Figure 6).

TNKS were previously implicated in the regulation of cell migration. Since TNKS are overall overexpressed in several cancer types and are promising targets to block the Wnt pathway in particular, the pharmacological inhibition of TNKS or their siRNA-mediated depletion was tested and shown in numerous studies to decrease the migration and invasion of cancer cell lines [34,35,36,37,38,39,40,41]. Given the plethora of TNKS partners, the mechanisms at play may not be the same in all cell systems. One TNKS partner, TNKS1BP1, which is PARylated, negatively regulates cancer cell invasion by interacting with the capping protein and decreasing the actin filament dynamics [42]. TNKS1BP1 is downregulated in pancreatic cancer.

Here, we report that TNKS aggregates possess the properties of biomolecular condensates and that Arpin dissolves these condensates upon overexpression. Biomolecular condensates correspond to liquid–liquid phase separation due to multimeric proteins and multimeric ligands [43]. TNKS possess multiple ARCs and oligomerize through their SAM motif [27,28]. The presence of multivalent ligands induces TNKS condensation [44]. On the contrary, Arpin is a monomeric protein with a single TNKS-binding site that fits very well in the consensus motif defined by the peptide display library [10]. Overexpressed Arpin is thus likely to saturate functional ARCs of TNKS, resulting in the displacement of endogenous multivalent ligands and the subsequent dissolution of TNKS condensates. However, at the endogenous levels of expression, we did not detect a role for Arpin either in TNKS condensation or in the efficiency with which TNKS regulate their substrates. Therefore, it is still unclear at this point how Arpin controls TNKS and how TNKS control the migration persistence.

Biomolecular condensation might play a role in tumor cells where TNKS are overexpressed. However, it should be stressed that Arpin is, on the contrary, downregulated in tumors compared to normal adjacent tissues [45,46,47,48]. Since the Arpin and TNKS levels vary in opposite directions in cancers, the interaction between Arpin and TNKS reported here might be more important in the regulation of cell migration in untransformed cells than in tumor cells. In untransformed cells, such as MCF10A cells, Arpin appears to control the migration persistence through a two-pronged mechanism, involving the independent binding of Arp2/3 or TNKS to the same binding site of Arpin.

## 4. Materials and Methods

### 4.1. Plasmids, gRNAs, and Transfection

For expression in 293 Flp-In cells, human Arpin ORF was cloned in pcDNA5 FRT His PC TEV Blue between the FseI and AscI sites. The 293 Flp-In stable cell line expressing PC Arpin was obtained as previously described [49]. For the yeast two-hybrid screening, full-length human Arpin was cloned into pB27 in fusion with the LexA DNA-binding domain. A random primed cDNA library from human placenta was screened by Hybrigenics using a mating protocol and 2 mM 3-aminotriazole to reduce the background. Arpin G8A, G189A, G218A, triple G8A-G189A-G218A, R213A, R213D, G218D, and W224A mutants were obtained in pcDNA5 His PC TEV Arpin plasmid using the QuikChange Lightning Site-Directed Mutagenesis Kit (Agilent Technologies, Santa Clara, CA, USA). ORFs encoding Arpin WT and mutant forms were subcloned between the FseI and AscI sites into custom-made pcDNAm FRT PC GFP, MXS PGK ZeoM bGHpA EF1Flag mScarlet Blue2 SV40pA, and MXS EF1Flag Blue2 SV40pA PGK Blasti bGHpA plasmids.

The 293T cells were transfected using Calcium Phosphate or Lipofectamine 3000 (Thermo Fisher Scientific, Waltham, MA, USA). The 293T *ARPIN* KO cell line (clone #44) was generated with the CRISPR/Cas9 system, as previously described for MCF10A cells [32]. Stable MCF10A cells expressing Flag-Arpin WT, Flag-Arpin G218A, and Flag-Arpin W224A were obtained in MCF10A *ARPIN* KO cells [32] by transfecting with the custom-made MXS plasmids described above using Lipofectamine 3000 (Thermo Fisher Scientific). Cells were selected with 10 µg/mL of Blasticidin (InvivoGen, San Diego, CA, USA). Individual clones were picked with cloning rings, and Flag-Arpin expression was verified by Western blotting. MCF10A Arpin KI cell lines were generated with the CRISPR/Cas9 system. The following targeting sequences were used: 5′-TCCCGACCGCCCGGGCACCC-3′ targets before the ATG codon in exon1 and 5′-GATTTCTCTAGGATGACTGA-3′ targets after the Stop codon in exon6 of Arpin. These sequences were flanked by the BbsI restriction site. Corresponding oligonucleotides were annealed and cloned in the pX330 plasmid expressing human SpCas9 protein (Addgene #42230). The donor plasmids were constructed as follows. The sequences were amplified by PCR with Phusion polymerase (Thermo Fisher Scientific): the Arpin homology arm right (HR) flanking Cas9-targeted site was amplified from genomic DNA extracted from wild-type MCF10A cells (NucleoSpin tissue extraction kit, Macherey-Nagel, Düren, Germany), and Puro-T2A were amplified from the custom-made plasmid MXS Puro bGHpA using primers containing the T2A sequence. Amplified sequences were checked by Sanger sequencing. Arpin homology arm left (HL) was synthesized by Eurofins. The donor cassette was constructed by assembling HL, Puro-T2A, GFP-Blue2, and HR by MXS-Chaining [50]. Full-length ORFs encoding Arpin WT, G218A, G218D, and W224A were then subcloned in the constructed donor plasmid between the FseI and AscI sites. MCF10A *ARPIN* KO cells were transfected with the Cas9- and gRNA-containing pX330 plasmid and the donor plasmids described above using Lipofectamine 3000 (Thermo Fisher Scientific). Cells were selected with 0.5 µg/mL of puromycin (InvivoGen). Single clones were picked with cloning rings, expanded, and analyzed by Western blotting.

### 4.2. Cell Culture and Drugs

MCF10A cells were maintained in DMEM/F12 medium (Thermo Fisher Scientific) supplemented with 5% horse serum (Sigma, Hongkong, China), 100 ng/mL cholera toxin (Sigma), 20 ng/mL epidermal growth factor (Sigma), 0.01-mg/mL insulin (Sigma), 500 ng/mL hydrocortisone (Sigma), and 100 U/mL penicillin/streptomycin (Thermo Fisher Scientific). The 293, 293T, MEF, and HeLa cells were maintained in DMEM medium (Thermo Fisher Scientific) supplemented with 10% fetal bovine serum (Thermo Fisher Scientific) and 100-U/mL penicillin/streptomycin (Thermo Fisher Scientific). Cells were incubated at 37 °C in 5% CO_2_. All cells and stable clones were routinely tested for mycoplasma and found to be negative. The TNKS inhibitor XAV939 was from Sigma. The 293T parental and TNKS double-KO cell lines were kindly provided by Dr. S. Smith, Skirball Institute, New York School of Medicine.

### 4.3. Immunoprecipitation and GST Pulldown

The 293 cells stably expressing His-PC-Arpin or the empty plasmid as a control (Figure 1) were lysed in 50 mM Hepes, pH 7.7, 150 mM NaCl, 1 mM CaCl2, 1% NP40, 0.5% Na Deoxycholate, and 0.1% SDS 1% supplemented with a protease inhibitor cocktail (Roche, Basel, Switzerland). Clarified lysates were incubated with 10 µL of HPC4-coupled beads (Sigma) for 3 h at 4 °C. After 5 washes in the same buffer, beads were analyzed by SDS-PAGE. For TNKS identification, tryptic peptides were analyzed by Nano-LC-MS/MS analyses using a LTQ Orbitrap Velos mass spectrometer (Thermo Scientific) coupled to the EASY nLC II high-performance liquid chromatography system (Proxeon, Thermo Scientific). Peptide separation was performed on a reverse-phase C18 column (Nikkyo Technos, Tokyo, Japan). Nano-LC-MS/MS experiments were conducted in a data-dependent acquisition method by selecting the 20 most intense precursors for CID fragmentation and analysis in the LTQ. Data were processed with Proteome Discoverer 1.3 software, and protein identification was performed using the Swiss-Prot database and MASCOT search engine (Matrix Science, Merrillville, IN, USA).

For PC immunoprecipitation of PC-Arpin WT or mutants (Figure 2), 293T cell lysates were lysed in 50 mM KCl, 10 mM Hepes, pH 7.7, 1 mM MgCl_2_, 1 mM EGTA, and 1% Triton TX100 supplemented with a protease inhibitor cocktail (Roche). Twenty microliters of HPC4 beads were supplied with 1 mM Ca^2+^, protease inhibitor cocktail (Roche), and 2 µM ADP HDP PARG inhibitor (Merck Millipore, Burlington, MA, USA). Beads were incubated with extracts for 1 h at 4 °C, washed 5 times in the same buffer, and analyzed by Western blotting. For GFP immunoprecipitation (Figure 2), 293T cells transiently transfected with GFP-Arpin were lysed in 50 mM KCl, 10 mM Hepes, pH 7.7, 1 mM MgCl_2_, 1 mM EGTA, and 1% Triton TX100 supplemented with a protease inhibitor cocktail (Roche). Extracts were incubated with anti-GFP agarose beads (GFP-trap, Chromotek, Planegg, Germany) for 2 h at 4 °C, washed 5 times in the same buffer, and analyzed by Western blotting.

For the immunoprecipitation of endogenous Arpin, 293T cells were lysed in 50 mM KCl, 10 mM Hepes, pH 7.7, 1 mM MgCl_2_, 1 mM EGTA, and 1% Triton TX100 supplemented with a protease inhibitor cocktail (Roche). Clarified extracts were incubated for 2 h with agarose beads previously coupled to 10 µg of nonimmune rabbit IgG or 10 µg of affinity-purified Arpin antibodies (according to the manufacturer’s protocols; AminoLink Coupling Resin, Thermo Fisher Scientific). Beads were incubated with extracts for 2 h at 4 °C, washed 5 times in the same buffer, and analyzed by Western blotting. HeLa cell pellets were lysed in 50 mM Hepes, pH7.7, 150 mM NaCl, 1 mM CaCl2, 1% NP40, 0.5% Na Deoxycholate, and 0.1% SDS 1% supplemented with a protease inhibitor cocktail (Roche). Twenty micrograms of GST fusion protein, and 20 µL of Glutathione Sepharose 4B Beads (GE Healthcare, Chicago, IL, USA) were incubated with 1 mL of HeLa cell extract for 2 h at 4 °C. When indicated, purified ARC4 protein is added into the mixture to complete the interaction. Beads were washed 5 times in the same buffer and analyzed by Western blotting.

### 4.4. SEC-MALS

For SEC-MALS, purified proteins were separated in a 15-mL KW-803 column (Shodex, Shanghai, China) run on a Shimadzu HPLC system. MALS, QELS, and RI measurements were achieved with a MiniDawn Treos, a WyattQELS, and an Optilab T-rEX (all from Wyatt Technology, Santa Barbara, CA, USA), respectively. Mass calculations were performed with ASTRA VI software (Wyatt Technology) using a dn/dc value of 0.183 mL·g^−1^.

### 4.5. Protein Purification and Analysis of Arpin Expression

Arpin was cloned into the pET28 vector and produced as a N-terminal 6xHis-tagged protein. Expression was done in *E. coli* Rosetta(DE3)pLysS by the addition of 0.5 mM IPTG followed by overnight incubation at 25 °C. The bacterial pellet was resuspended in 50 mM Tris, pH 8.0, 300 mM NaCl, 10 mM imidazole, 0.25-mg/mL lysozyme, and EDTA-free protease inhibitor cocktail (Roche). Lysis was carried out by sonication. Arpin was purified from the soluble fraction by Ni2+-affinity chromatography (HisTrap HP, GE Healthcare) using 125 mM imidazole for elution, followed by size exclusion chromatography on a Superdex 75 16/60 HL column (GE Healthcare) equilibrated in 20 mM Hepes, pH 8.0, and 100 mM KCl. ARC4 from TNKS2 was expressed and purified as already described [5]. For pulldown assays, the WWE domain of RNF146 (amino acids 100–175) and Arpin were cloned in a modified pGEX vector containing a TEV protease cleavage site after the GST moiety that was not used here. GST fusion proteins were purified on glutathione-sepharose beads using a standard protocol. Purity was followed by SDS-PAGE, and the concentration of purified proteins was estimated by UV spectrophotometry using theoretical extinction coefficients at 280 nm.

### 4.6. SDS-PAGE, Western Blots, and Antibodies

For analysis of the Arpin expression, MCF10A cells were lysed in RIPA buffer (50 mM Hepes, pH 7.5, 150 mM NaCl, 1% NP-40, 0.5% DOC, 0.1% SDS, and 1 mM CaCl_2_) supplemented with the EDTA-free protease inhibitor cocktail (Roche), and the lysates were clarified and analyzed by Western blotting.

SDS-PAGE was performed using NuPAGE 4–12% Bis-Tris gels (Life Technologies, Carlsbad, CA, USA). For Western blots, the proteins were transferred using the iBlot system (Life Technologies, Invitrogen) and developed using HRP-coupled antibodies, a Supersignal kit (Pierce, Thermo Fisher Scientific), and a LAS-3000 imager (Fujifilm, Tokyo, Japan) or AP-coupled antibodies and NBT/BCIP as substrates (Promega, Madison, WI, USA).

Rabbit polyclonal antibodies obtained and purified using full-length Arpin were previously described [4]. The following commercial antibodies were used: TNKS1/2 pAb (H-350, Santa Cruz Biotechnology, Dallas, TX, USA), ArpC2 pAb (Millipore, Burlington, MA, USA), ArpC3 pAb (Sigma-Aldrich, St. Louis, MO, USA), α-Tubulin mAb (Sigma), Axin1 mAb (C76H11, Cell Signaling Technology, Danvers, MA, USA), and PTEN pAb (Cell Signaling Technology, #9552).

### 4.7. Immunofluorescence

For immunofluorescence, MCF10A cells were seeded on glass coverslips coated with 20 µg/mL bovine fibronectin (Sigma) for 1 h at 37 °C in PBS. Then, cells were fixed with 4% paraformaldehyde, permeabilized in 0.5% Triton X-100, and blocked in 0.1% Triton X-100 and 2% bovine serum albumin (BSA; Sigma) in PBS for 1 h at RT. Coverslips were incubated with the indicated primary antibodies for 1 h at RT, then with anti-rabbit IgG antibodies conjugated with Alexa Fluor 488 (Invitrogen, Carlsbad, CA, USA). Imaging was performed on an Axio Observer microscope (Zeiss, Jena, Germany) with the 63×/1.4 oil objective.

### 4.8. Live Confocal Imaging

GFP-TNKS2 condensates were imaged using a confocal laser scanning microscope (TCS SP8, Leica, Wetzlar, Germany) equipped with an inverted frame (Leica), a high NA oil immersion objective (HC PL APO 63×/1.40, Leica), and a white light laser (WLL; Leica). The acquisition was performed using LASX software. To capture condensate fusion events, GFP-TNKS2 expressing cells were imaged every 0.79 s for 158 s. For the FRAP experiment, single GFP-TNKS2 condensates were bleached during 1.5 s and imaged every 5 s for 105 s. The aggregate intensity was manually normalized to the background, and the recovery curve presents the mean intensity and standard deviation (SD) (*n* = 30 condensates from 16 cells). To measure the fraction of cells with detectable condensates (*n* > 1), MCF10A cells co-expressing GFP-TNKS2 and mScarlet, mScarlet-Arpin WT, or the G218D mutant were imaged (data from 3 independent experiments with at least 115 cells analyzed for each condition are represented). The total quantity of GFP-TNKS2 aggregates per cell was measured in 3 independent experiments. Images were analyzed in ImageJ as follows: first, the MaxEntropy threshold was applied, and then, the number of condensates was counted with the Analyse Particles function (size > 0.15 µm; measures from 60 cells were pooled).

### 4.9. Live Imaging and Analysis of Cell Migration and Statistics

MCF10A cells were seeded onto glass-bottomed µ-Slide (Ibidi, Gräfelfing, Germany) coated with 20 µg/mL of fibronectin (Sigma). Imaging was performed on an Axio Observer microscope (Zeiss) equipped with a Plan-Apochromat 10×/0.25 air objective; the Hamamatsu camera C10600 OrcaR2; and the Pecon Zeiss incubator XL multi S1 RED LS (Heating Unit XL S, Temp module, CO_2_ module, Heating Insert PS, and CO_2_ cover). Images were acquired every 10 min for 24 h. Single-cell trajectories were obtained by tracking cells with ImageJ and analyzed using DiPer software [33] to obtain the migration parameters: directional autocorrelation, mean square displacement, average cell speed, and single-cell trajectories plotted at the origin. Data from two (Flag-Arpin cell lines) or three (GFP-Arpin knock-in cell lines) independent experiments were pooled for the analysis and plotted. The results are expressed as the means and standard errors of the mean (s.e.m). The average cell speed was analyzed using GraphPad software with the Kruskal–Wallis test. For migration persistence, a statistical analysis was performed using R. Persistence, measured as the movement autocorrelation over time was fit for each cell by an exponential decay with a plateau (as described in reference [51]):$$A=\left(1-A_{min}\right)*e^{-\frac{t}{\tau }}+A_{min}$$
where *A* is the autocorrelation, *t* the time interval, *A_min_* the plateau, and *τ* the time constant of decay. The plateau value *A_min_* is set to zero for the cell lines in vitro, as they do not display overall directional movement. The time constant *τ* of the exponential fits were then plotted and compared using the Kruskal–Wallis test. Four levels of statistical significance were defined: * *p* < 0.05, ** *p* < 0.01, *** *p* < 0.001, and **** *p* < 0.0001.

## Figures and Tables

**Figure 1 ijms-22-04115-f001:**
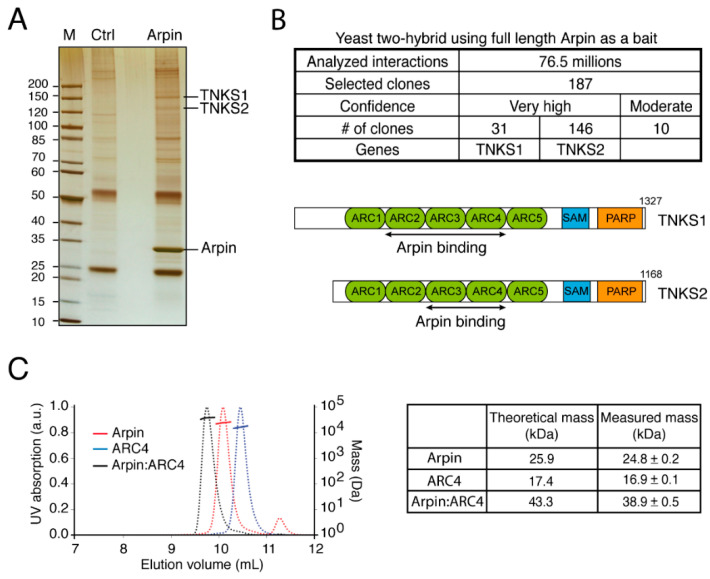
Tankyrases are major Arpin-interacting partners. (**A**) A 293-cell line stably expressing tagged Arpin was used to immunoprecipitate Arpin and its associated proteins (silver staining). Proteins were identified by mass spectrometry. (**B**) Full-length Arpin was used as bait in a yeast two-hybrid screening. The retrieved clones of TNKS were mapped to regions indicated with black arrows within the region containing Ankyrin Repeat Clusters (ARCs). (**C**) The molecular species obtained by mixing purified Arpin and purified ARC4 of TNKS were analyzed by SEC-MALS.

**Figure 2 ijms-22-04115-f002:**
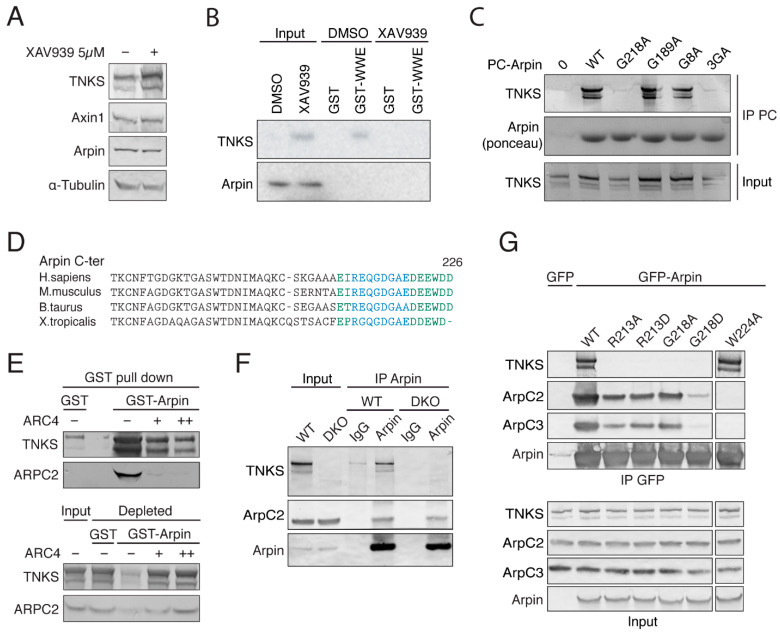
The Arpin–TNKS interaction does not appear to regulate the Arpin levels and requires a C-terminal consensus motif for TNKS binding. This consensus motif overlaps with the Arp2/3 interaction site, but TNKS does not modulate the Arpin–Arp2/3 interaction in cells. (**A**) The Arpin–TNKS interaction does not appear to regulate the Arpin levels. The 293T cells were treated with the TNKS inhibitor XAV939 or the vehicle for 24 h, and the lysates were analyzed by Western blots. (**B**) MEF lysates were subjected to GST pulldown using the WWE domain of E3 ligase RNF146, which recognizes PARylated proteins. XAV939 (1 µM) was used to block TNKS catalytic activity. (**C**) The mapping of TNKS binding to Arpin. PC-tagged Arpin WT, G8A, G189A, G218A, and a triple-mutant G8A-G189A-G218A were transiently expressed in 293T cells. Anti-PC agarose beads were used to immunoprecipitate tagged Arpin. (**D**) TNKS (blue) and Arp2/3 (green)-binding sites overlap in the C-terminus acidic tail of Arpin. (**E**) TNKS binding competes with Arp2/3 binding in vitro. MEF lysates were incubated with purified GST or GST-Arpin immobilized on glutathione beads. Increasing concentrations of purified ARC4 were added to the lysate as indicated. GST beads, input lysate, and depleted lysates were analyzed by Western blots. (**F**) Lysates from WT and TNKS double-KO 293T cells were analyzed by Arpin immunoprecipitations with non-immune IgG as the control. (**G**) GFP-Arpin WT, R213A, R213D, G218A, G218D, and W224A were transiently expressed in 293T cells and immunoprecipitated.

**Figure 3 ijms-22-04115-f003:**
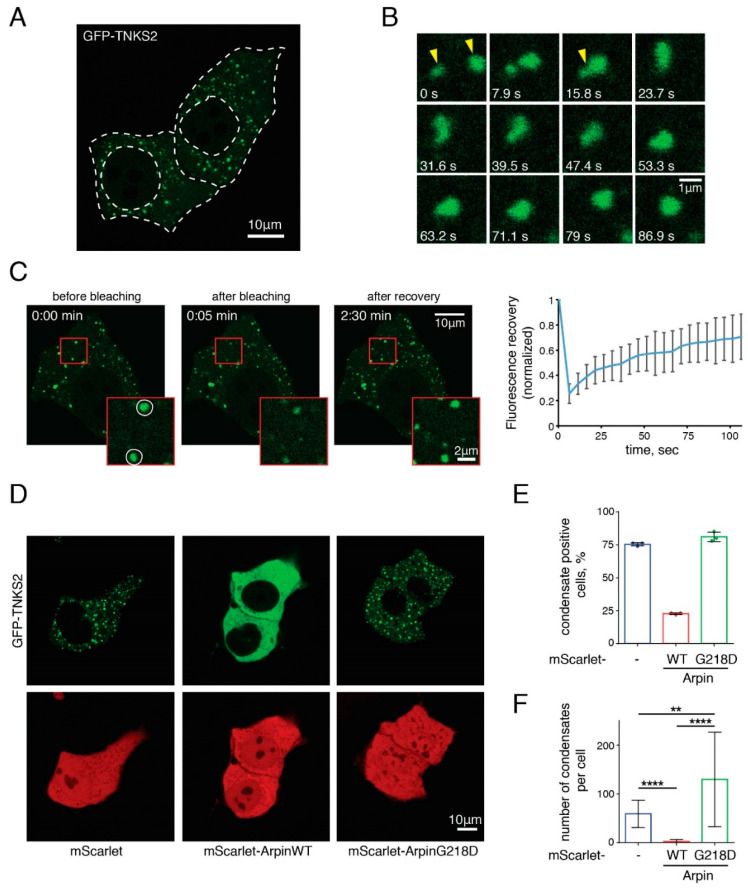
Arpin regulates the ability of TNKS to form biomolecular condensates. (**A**) GFP-TNKS2 was transiently expressed in MCF10A cells. White, dashed lines delimit the cell membranes and nuclei. (**B**) The fusion of two GFP-TNKS2 aggregates (yellow arrows) was imaged in a time-lapse series. (**C**) GFP-TNKS2 aggregates were analyzed by FRAP (bleached areas are indicated by white circles in the magnified image boxed in red). (**D**) The wild type or the G218D Arpin fused to mScarlet were co-expressed with GFP-TNKS2 in the MCF10A cells. (**E**) The quantification of the proportion of transfected cells that display TNKS2 condensates (3 biological replicates). (**F**) The quantification of the number TNKS2 condensates per cell (Kruskal–Wallis test, ** *p* < 0.01 and **** *p* < 0.0001).

**Figure 4 ijms-22-04115-f004:**
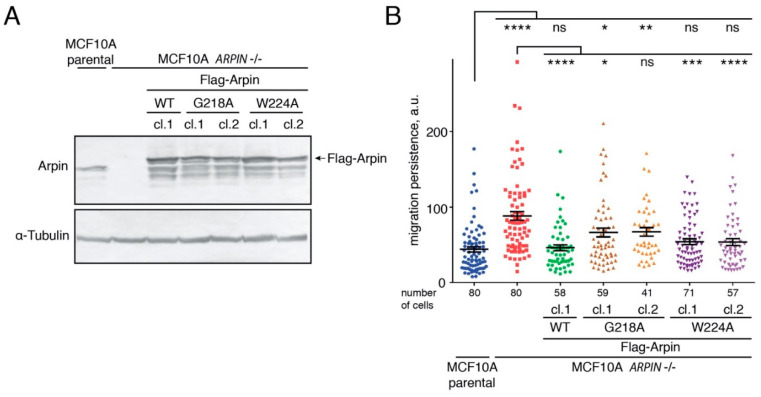
The Arpin–TNKS interaction is important for the control of cell migration persistence. (**A**) Stable clones expressing Flag-Arpin WT, Flag-Arpin G218A, and Flag-Arpin W224A after plasmid random integrations were generated from the *ARPIN* KO MCF10A cell line. Endogenous and exogenous Arpins were revealed by Western blotting using Arpin antibodies. (**B**) The random migration of single cells was tracked for 7 h, and the migration persistence was extracted from the trajectories (Kruskal–Wallis test, * *p* < 0.05, ** *p* < 0.01, *** *p* < 0.001, and **** *p* < 0.0001; the number of cells ranged from 41 to 80).

**Figure 5 ijms-22-04115-f005:**
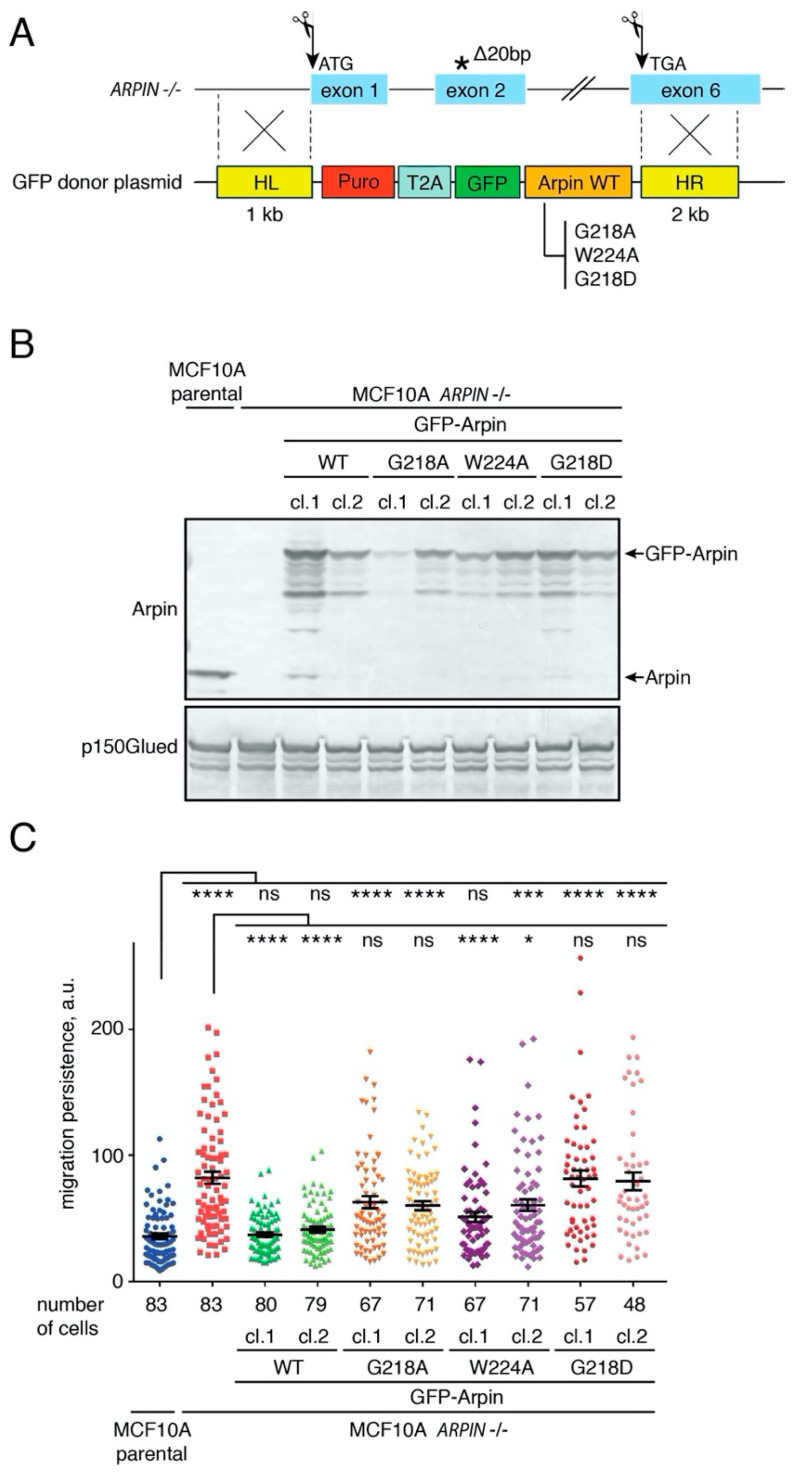
The Arpin–TNKS interaction controls cell migration persistence. (**A**) A scheme of the knock-in strategy used to rescue *ARPIN* KO cells. Two gRNAs allowing the excision of the whole Arpin locus were designed. The premature stop codon that appears due to a 20-bp deletion is indicated by an asterisk. Homology-directed repair was used to integrate selectable donor cassettes. The cassettes contained a single Open Reading Frame encoding the puromycin resistance gene, the viral self-cleaving T2A peptide, GFP and Arpin WT, G218A, W224A, or G218D. (**B**) The expression of GFP-Arpin WT and mutant forms in selected cell lines was assessed by Western blotting using anti-Arpin antibodies. (**C**) The migration persistence was extracted from the trajectories of randomly migrating single cells (Kruskal–Wallis test, * *p* < 0.05, *** *p* < 0.001, and **** *p* < 0.0001; the number of cells ranged from 47 to 83).

**Figure 6 ijms-22-04115-f006:**
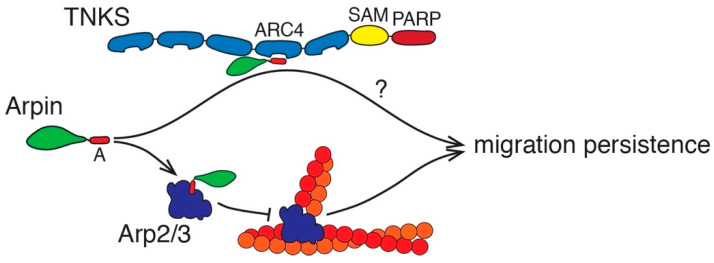
Working model. Arpin controls migration persistence through two pathways. The acidic tail (A) of Arpin binds to Arp2/3 and to Tankyrases (TNKS). TNKS are modular proteins composed of 5 Ankyrin Repeat Clusters (ARC), an oligomerization domain (Sterile Alpha Motif; SAM), and a catalytic domain (Poly ADP Ribosyl Polymerase; PARP). Arpin binds to ARC. Arpin, as a monomeric ligand of ARC, prevents the biomolecular condensation of TNKS. The mechanism, whereby the Arpin–TNKS interaction regulates the migration persistence, remains to be established.

## Data Availability

Not applicable.

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
