# Peer review of "Arpin Regulates Migration Persistence by Interacting with Both Tankyrases and the Arp2/3 Complex"

_ijms, 2021, doi:10.3390/ijms22084115_

Round 1

Reviewer 1 Report

This study describes that the binding partners of arpin, a Arp2/3 inhibitory protein is TNKS 1 and 2 in vitro and this is also the case in vivo. In addition, They found  its binding site to TNKS and Arp2/3, and they assessed the effects of its mutation. Only the mutation that affects both interactions rendered arpin completely inactive, suggesting the existence of two independent pathways, by which arpin controls migration persistence. Together these data suggest that TNKS might be mediating the function of Arpin rather than regulating arpin. I found the experiment and analysis to be mostly sound, with minor comments:

1. page 5 TNKS form an oligomer. Do TNKS1 and TNKS2 form a hetero-oligomer together? Does the aggregate contain both TNKS1 and TNKS2?

2. When the cell migration persistence changes, do the cells change their morphology?  Or the orientation or frequencies of pseudopods extensions are changed?

3. Figure 5   In the mutants, does the amount of actin or arp2/3 changed?

4. Please, add a model schema for the relationship among molecules arpin, arp2/3, and TNKS as a final figure, which is easy for the readers to understand .

5. As a negative regulators for Arp2/3, GMF and coronin are also known. Please discuss the relationship among them including arpin.

Author Response

see attached document

Reviewer 2 Report

Review

Simanov identified TNKS as new interaction partner of Arpin and analyzed the role of this protein-protein interaction for migration. They found that TNKS binds to the C-terminal tail of Arpin, where also the binding site for Arp2/3 is located. Thus TNKS and Arp2/3 bind the same site in the Arpin molecule and at least in vitro excess of TNKS can displace Arp2/3 from Arpin. However, in TNKS knock out cells the concentration of Arp2/3 did not increase, indicating that in cells TNKS does not compete for Arp2/3. In addition, the authors show that TNKS form aggregates when overexpressed as GFP-fusion protein and reveal that both, Arp2/3 and TNKS are important for Arpin-mediated inhibition of cellular migration.

This is a very interesting, well-conducted study and the authors performed a lot of controls to validate their results.

However, some minor revisions are necessary:

  • It is absolutely required to show Coomassie-stained gels from protein purification or enrichment. In addition, the methods how the proteins were expressed and purified has to be described in detail in the method section.
  • Aggregation overexpressed GFP-TNKS could be simply an overexpressing artifact. Since this result is not important for the story, it should be deleted from the manuscript.
  • Please describe in the result section that the “KO cells” were generated by CRISPR-CAS and the method used to analyze cellular migration.
  • Finally, the mechanism how TNKS controls the effect of Arpin on cell migration is still unclear. In the discussion section, future research should be suggested to address this problem.
